# Research on Rural Landscape Preference Based on TikTok Short Video Content and User Comments

**DOI:** 10.3390/ijerph191610115

**Published:** 2022-08-16

**Authors:** Hao Chen, Min Wang, Zhen Zhang

**Affiliations:** School of Economics and Management, Anhui Agricultural University, Hefei 230036, China

**Keywords:** TikTok short video, content analysis, user comments, rural landscape preference

## Abstract

Landscape is the visual embodiment of the human–environment relationship. It is an important field for understanding and shaping the relationship between human society and the environment, and it is also the focus of multidisciplinary attention. Rural landscape construction is of great significance to the development of rural tourism and rural revitalization. The results and preferences from landscape evaluation are an important basis for landscape construction. This paper selected 222 rural landscape short video works published on the TikTok short video platform; extracted the basic elements of rural landscapes using video content analysis and according to grounded theory; condensed 32 basic categories and 12 main categories; and formed a rural landscape system composed of three core categories: rural ecological, living, and productive landscapes. The short video user comment data were mined using ROST CM6 software, to analyze the rural landscape preferences of video viewers. The results showed that the short video users had a high overall evaluation of rural landscapes, but there were differences among the three core rural landscape preference categories. Users had a high preference for the architectural landscape and ecological landscape in the rural lifestyle landscape but raised concerns about the impact of disharmonious infrastructure and service facilities, existing security risks, and environmental health on the rural landscape. This paper suggests that we should pay attention to the construction of rural artistic conceptions and the expression of nostalgia, enhance experiences to enhance perception, and strengthen the protection of natural and cultural landscapes.

## 1. Introduction

Landscape is the visual aspect of the human–environment relationship, and has both natural and cultural attributes [1]. Landscape is an important field for understanding and shaping the relationship between human society and the environment, and it is also an important means of providing landscape services and achieving human well-being [2]. Under pressure from grassroots mass movements, the economic development brought about by the destruction of ecologies and the urban environment has been protested against [3], and European and American countries have successively issued relevant laws and regulations to protect valuable traditional landscapes and regulate landscape construction activities. Landscape has gradually become the focus of multidisciplinary attention. Modern landscape research first focused on cities, and then gradually expanded to rural areas [4]. Rural landscapes are a comprehensive manifestation of economic, cultural, social, natural, and other phenomena in rural areas. Rural landscape construction is of great significance to the development of rural tourism and the implementation of China’s rural revitalization strategy. From the perspective of rural sociology and landscape design, agricultural production behavior, residents’ behavior, and the practice of rural landscape design constitute the basis of rural landscapes. The development of rural tourism pays attention to what the tourists see, enhancing the rural landscape’s natural beauty and visual effect. Against the background of globalization, informatization, and urbanization, changes in landscape scenes are more intense than before [5,6]. However, while rural modernization construction promotes rural changes and reconstruction [7], the rural landscape also encounters the phenomenon of alienation. Tourism development in some regions has caused cultural changes [8] and memory loss [9], which may lead to the annihilation of local landscapes. Urbanization, homogenization, and damage to the landscape affect the sustainability of rural landscapes.

Relevant research at home and abroad has mainly involved the concepts [10,11,12,13], characteristics [14,15], types [16,17,18], composition [16,17,19], value [20,21,22,23,24], preferences [25,26,27], evaluation [11,19,25,26,28], land use [19,28], planning and design [11,29], and management [12,29] of rural landscapes and their relationship with the development of rural tourism [12,18] and the construction of beautiful villages [30]. Among these, landscape value is the field that has received the most attention [2,20,21,22], and landscape value analysis provides a way to understand viewers’ attitudes toward rural landscapes [23]. The thirteen basic landscape value types proposed by Brown (2000), namely aesthetic, economic, leisure, life-sustaining, biodiversity, learning, spirit, immanence, history, future, survival, treatment, and culture, are widely used [24]. Rural landscape preference belongs to the field of landscape value assessment, and preference can be understood as the interaction of each visible feature of the landscape with the relevant psychological processes (perception, cognition, and emotion) in the viewers’ perception [31]. Relevant studies have shown that, in addition to being influenced by the attributes of the landscape itself, rural landscape preferences are also influenced by the viewer’s socioeconomic status, education level, environmental attitude, entertainment motivation, and other factors [32]. Due to the different subjective criteria of viewers in evaluating rural landscapes [33], there are significant individual and group differences in landscape preferences [34,35]. For example, tourists tend to appreciate agricultural landscapes more than permanent residents, because tourists perceive the difference between agricultural landscapes and urban landscapes [36]. Farmers generally pay more attention to different landscape attributes than the general public, because farmers mainly perceive the landscape through the benefits produced by the land [37], but farmers have a weaker perception of landscape changes [38]. Residents’ attachment to the local rural landscape is the main motivation for residents to participate in rural management and conservation [39]. Landscape professionals judge landscapes based more on their own knowledge than on perceived beauty [19]. Although there are large individual differences in evaluation results, there is a high degree of consistency in the preferences of rural landscape aesthetics [40].

The evaluation of landscape visual preference in related research includes direct and indirect methods. The direct method is a direct analysis of the landscape viewers’ evaluation of the landscape. The direct method can display the positive and negative impacts of the landscape on viewers through the satisfaction and pleasure of the viewers, but the results are often influenced by the characteristics of the sample [41,42,43]. The indirect method is to construct a landscape element index system for a certain type or region of landscape. An overall evaluation is obtained through the aggregation of the index analysis results, but this method is insufficient in terms of the interaction effects between elements [19] and the subjective judgment of the viewer. Landscape preference research mostly observes audience responses to visual stimuli, and the sources of landscape materials usually include GIS technology extractions [44,45,46,47], VR panoramic displays [48], and social media photos [49,50,51]. Rural landscape photos are most commonly used as stimuli. Although some experts note that photos cannot fully reflect the actual scene and multisensory experience, such as sound and taste, this method is still widely used, because of its simple operation and low cost. Relevant data analysis usually includes the Kruskal–Wallis test, hierarchical Bayesian analysis, neurophysiological analysis of viewers viewing photos, etc. The results of an analysis mostly reflect the audience’s preference for different types of rural landscape. For example, the importance of perceived visual quality decreases in relationship to increasing landscape barrenness, well-preserved artificial elements, vegetation coverage, water volume, mountains, and color contrast [19]. There are both positive sentiments, such as those about vegetation and water in rural landscapes, and negative sentiments about the visual quality of negative artificial elements, such as roads, alterations, and dilapidated buildings [19]. The results of landscape preference evaluation can be considered as factors for different subjects when making landscape construction decisions. This paper intends to analyze users’ landscape preferences using the content of TikTok short videos. Compared to landscape photos, short videos express richer landscapes and present more dynamic effects.

## 2. Data Sources and Sample Analysis

Since its launch in 2016, the TikTok short video platform has developed rapidly and has become an important distribution center for various types of information, due to its intuitive and popular methods of knowledge dissemination. If a video resource can arouse an emotional resonance, the user will give a “like” to express their emotion, which can be happy, supportive, nostalgic, or sad, and a higher degree of emotional resonance will also cause users to forward the video resource. In this study, the TikTok short video platform was selected to conduct an on-site search through keywords such as “rural landscape” and “rural scene” and to extract highly approved video resources with more than 10,000 likes. The videos were released during a two-year period, from October 2019 to September 2021. Excluding the repeated videos, a total of 222 valid short videos were obtained, and these short videos received a total of 35.21 million likes and 1.56 million forwards or shares. TikTok users, 55% of whom are women, are predominantly young, with 49% of users being young people aged 18–35 (data from “TikTok User Portrait Analysis, Q1 2022”).

Short video content production can be divided into two modes: professional generated content (PGC) and user generated content (UGC). UGC has a high degree of originality and is the mainstream creation mode of TikTok short videos. It also conforms to the characteristics of a short video duration and low production threshold. Among the rural landscape short videos used in this study, UGC accounted for 81.1% and PGC accounted for 18.9%. All the images in the visual elements are clear and stable, with few animations effects and subtitles. The auditory elements are dominated by music and contemporaneous sounds. There are a large number of videos with local positioning, and the main purpose of this is to share high-quality rural landscapes, rather than to publicize scenic spots. The presentation method is mainly full screen, which is in line with the user habit of watching videos in full screen. The length of the videos is mainly within 30 s, which is in line with the “short” characteristics of short videos. The main method of video interaction is likes, followed by comments and sharing (Table 1).

## 3. Analysis of Rural Landscape Elements Based on Short Video Content

As a qualitative research method, the grounded theory advocates the development of theory in data-based research. Its core is the continuous comparison of concepts and categories, which requires the collection of objective data and information that is authentic, comprehensive, and representative from the objects, avoiding the limitations of predetermined theoretical models or empirical concepts under the empirical research paradigm. Therefore, the adoption of grounded theory can make the conclusions more real, comprehensive, and accurate [52]. Based on these reasons, this paper uses grounded theory to explore rural landscape types and viewer preferences.

### 3.1. Landscape Description and Open Coding

Unlike text information, and aside from the short text content published along with the short video, the majority of the information is presented in the form of pictures, music, commentary, etc., and the relevant content needs to be “interpreted”. This study followed the principle of objectivity. According to the video content, members of the research team identified and recorded the landscape information in the short videos and completed the extraction of landscape elements from 222 short videos, to form a landscape element description text of 25,876 words. Among them, the static landscape elements included “firewood heaps, wooden fences, green plants, stone steps, potted plants, wooden tables, tea sets, bamboo hats”, dynamic activity elements, such as “children herding cattle, cooking, cattle farming, planting rice seedlings, washing vegetables”, and sound elements, such as “dog barking, water flowing, bird chirping, insects chirping”. Open coding was carried out synchronously with the landscape description. Utilizing high-frequency vocabulary analysis of the landscape description text (Figure 1), 184 high-frequency words were obtained. Based on these words, the preliminary category extraction was carried out, the extraction results were further compared with the description text to identify omissions, and 32 preliminary categories were formed, after several iterations of comparative analysis (Table 2).

### 3.2. Axial Coding

The basic categories extracted by open coding have common characteristics. According to the interrelationship and logical relationship between the basic categories, the main categories of landscape elements were further abstracted, classified, and refined. Combined with the classification of rural landscapes in existing research results, 12 main categories were formed, including astronomical phenomena and the weather, biological landscape, topography, rural residents, living scenes, facility supplies, food and clothing, infrastructure, rural buildings, production tools, production scenes, and production content.

### 3.3. Selective Coding

The 12 main categories formed by the axial coding were further refined, the common attributes among the main categories were analyzed, and the core categories were further extracted. The “production–living–ecological” concept is the most widely used rural composition dimension in the field of rural research [53], and it can comprehensively summarize all the constituent elements of the countryside. The 12 main categories extracted in this study have obvious “production–living–ecological” attributes. Therefore, the rural ecological landscape, rural living landscape, and rural productive landscape were taken as the core categories to construct the rural landscape element system (Figure 2). The results of the saturation test showed that no new concepts, categories, or relationships appeared in the 12 main categories and three core categories.

## 4. Rural Landscape Preference Analysis Based on User Comments

### 4.1. Data Mining and Cleaning

Semi-automatic data mining was carried out through a Python data collector and used the selenium function in Web automation tools to open the 222 video link addresses in the TikTok web version and automatically turn pages. All the content of user interactive responses was manually expanded, to obtain all comments and integrated into word text data, with a total of 6,073,251 words. As most video user comments were short, colloquial, and random, and as there were problems such as mixed, missing, or repeated content, the data were further cleaned. At the same time, incorrect words in the data were modified; meaningless data, such as username, garbled number, garbled English, and special symbols were eliminated; and the TXT data text in ANSI format was clarified.

### 4.2. High-Frequency Word Analysis

ROST CM6 software was used to analyze the rural landscape preferences of short video users, import the preprocessed text data into ROST CM6 software, delete stop words and perform word segmentation, extract words with two words or more, and retain the top 100 high-frequency words. By merging synonyms, eliminating duplicate values, and reducing the dimensionality of the data, to achieve the best clustering effect, we further removed meaningless high frequency words, such as “no longer”, “very”, “see”, and “good”, forming a group of high frequency words for rural landscape evaluation (Table 3) (only the top 50 words are listed).

The results show that the high-frequency characteristic words mainly reflect a positive evaluation of the rural landscape, not only in their praising description, nostalgia, and memory expression with the video landscape, but also in the emotional expression of yearning for the rural landscape environment represented by the video landscape. Words such as “picturesque scenery, small bridges flowing water, birds’ singing and flowers’ fragrance” reflect the beautiful rural artistic conception used to describe the rural landscape and express love; “fairyland on Earth, beautiful, wonderful, fine…” express praise for the rural landscape; “hometown, mother, grandmother, childhood…” express the good memories of hometown and missed relatives; “suitable, old-age care, yearning, travel…” express full of longing for the landscape and the artistic conception reflected in the video.

### 4.3. Semantic Network Analysis

The results of the semantic network analysis of ROST CM6 software were imported into Gephi software for visual expression of the semantic network of characteristic words. The network balance layout was formed by the interaction of forces between the characteristic words, which can display the strength of the semantic connection of the characteristic words in the network and provide an intuitive overall display for data analysis. The results show that two core semantic circles were formed (Figure 3). The first core semantic circle mainly expresses the nostalgic thoughts and memories of the short video viewers. Through the analysis of other characteristic words connected by the core characteristic word “childhood,” homesickness is the core emotional expression. These expressions include missing relatives in one’s hometown, such as a grandmother or mother; missing the rural scenery of one’s childhood, such as the old house they lived in as a child or the scene of rain; missing familiar sounds, such as insects singing, birds chirping, or dogs barking; and missing the happy and good times of childhood. The second core semantic circle takes “local” as the core characteristic of semantic words. This mainly reflects the recognition, praise, admiration, and association of short video viewers with the countryside. Due to this, it evokes memories of hometown and childhood life, arouses emotional resonance, is full of longing and yearning for one’s hometown, and generates the urge to return to the countryside.

### 4.4. Sentiment Analysis

Sentiment analysis of user comments in rural landscape short videos was conducted using ROST CM6 software. The results showed that positive emotions accounted for 46.96%, which was the highest; neutral emotions accounted for 40.70%; and negative emotions only accounted for 12.34% (Table 4). Short video users have a high level of appreciation of rural landscapes, and they use many words of praise, such as “fairyland on earth, picturesque scenery” and other words were used with high frequency, such as “indescribable beauty, it’s truly a fairyland on earth, it’s truly beautiful with this music”, which fully expresses the high admiration of short video users for this kind of video content. Although the proportion of neutral emotions was high, most of them were inquiries and associations about the beauty of the countryside, such as “Where is this, Guilin? Where is this scene taken?” Most of the negative emotions were concerns about rural infrastructure: “What if the power goes out at night when I live in this mountain?” Some users’ concerns may come from a lack of understanding of rural development and changes. The impact of electricity poles, wires, and sundries in the rural landscape on the landscape was also a source of negative emotions, but the proportion of intense negative emotions was low.

### 4.5. Short Video Users Landscape Category Preference Analysis

#### 4.5.1. Rural Ecological Landscape

The rural ecological landscape includes three main categories: rural astronomical phenomenon and the weather, topography, and biological landscape. This is the environment on which rural production and life rely and is the background factor for the formation of rural artistic conception. Harmonious rural ecological landscapes leave a deep impression on short video users, who highly praise the landscape. Landscape elements such as “rainy days, rivers, streams” accounted for a relatively high proportion of high frequency words for landscape elements, and words such as “air, scenery, picturesque scenery” were frequent in short video user comments. User comments mentioned: “There are mountains and water, which are so beautiful; I like the bamboo forest in the back; the beauty of the original ecology; What a picture of the countryside, which is beautiful and fresh.” Users associated the beautiful ecological landscape with beautiful poems, such as “Tangli fried snow and rain…; I have a pot of wine…”; One user even made a semi-creation based on this, “I have a pot of wine to drink between the mountains and rivers; the fishing boat is accompanied by birds, lucid waters with lush mountains; Tangli fried snow and rain, the smoke and the rain with thunder…”, all of which fully show the importance of green, natural, and harmonious rural ecological landscapes in the composition of rural landscapes. However, the fragility of the rural ecological landscape has led many viewers to put forward opinions, such as “there are weeds everywhere along the river; what to do with the flood; the water is too dangerous; how to deal with domestic water; how to discharge the sewage and what about the drinking water; how few trees; it is a pity that there are no peach trees; a few rooms with sparse trees; alas, we also cut down fruit trees and pine trees, and planted fast-growing eucalyptus”.

#### 4.5.2. Rural Living Landscape

The rural living landscape includes six main categories: rural residents, living scenes, facility supplies, food and clothing, infrastructure, and rural buildings. Rural life contains more rural history and culture, is the core embodiment of rural memory, and is an important component of rural cultural landscapes [53]. Among the high-frequency words of landscape elements, “brick houses, adobe houses, courtyards, fences, wooden houses…” are reflected in the architectural and cultural landscape of the countryside and are the landscape categories with the most visual impact in the rural living landscape. “Stone steps, stone roads, stone bridges…” are reflected in the rural infrastructure landscape, “smoke, washing clothes, washing vegetables, fishing, elderly individuals, women washing clothes, grandmother…” are reflected in rural life scenes, and the above categories account for the highest proportion of landscape elements and are the video viewers’ most recognized landscape elements. “Wooden stools, wooden tables, shelves, flavors, clothing” and other daily necessities, clothing, and dietary elements also account for a relatively high proportion. In short video user comments, words such as “house, fireworks, household, grandmother, mother” are frequent. In the user’s comments, it was mentioned that “it looks like the house in my hometown; where is it? Do you have a homestay? I truly want to go back to that water town in the south of the Yangtze River to see the house I have lived in and the road I have traveled; a tile house under the green bamboo forest; this kind of thing has been used when I was a child; good craftsmanship; how to buy this thing”, and so on. It can be seen that the rural life landscape triggers a home complex in the viewer [54], produces a “familiar” and “friendly” psychology, and generates the impulse to return to the countryside, which is the core meaning of the rural landscape. Owing to this, some viewers put forward concerns, such as “when it rains, the houses leak everywhere; there are a lot of mosquitoes; the beautiful scenery is destroyed by the house; now there are almost no houses with mud walls and small gray tiles in my hometown, which I miss; the gate and the house do not match; the stone used to harden roads, but now cement is used; the cement railing is perfect on the road from the gate to the creek; the roadside is overgrown with weeds”.

#### 4.5.3. Rural Productive Landscape

Production is the foundation of rural life and the core embodiment of farming culture [53]. The rural productive landscape includes five main categories: production tools, production scenes, production content, infrastructure, and rural buildings. In the high frequency words of landscape elements, the frequency of planting and breeding content such as “rice field, rapeseed, cattle, sheep” is relatively high. The visual effect portrays pastoral scenery as a production space. The interactive comments, including “there are fertile land, beautiful pond, mulberries and bamboos; the countryside is idyllic, comfortable and simple; I like this kind of green; this is the pastoral life I yearn for…” also reflect this feature. Production is more reflected in dynamic production activities, and those who have experience in production activities have a deeper feeling for the production and labor scenes, such as “dig bamboo shoots; reap rice with parents; only rural people know that this rice is hard-won; we can no longer see this kind of rural feeling; when it rains, parents will worry about the crops being flooded in the fields”, and other activities left a deep impression on them. However, “take a charger with you; it would be better if there were no telephone poles; the location of telephone poles affects the landscape a bit; the network is poor; I grew up in such an environment, which is very inconvenient…” also reflect the expectations of viewers about the rural production space.

## 5. Conclusions, Discussions, and Recommendations

### 5.1. Conclusions

(1) Through the screening of rural landscape short videos with many likes, the basic landscape elements in the short videos were extracted. According to grounded theory, the three core landscape categories of rural ecological landscapes, rural living landscapes, and rural productive landscapes were constructed. Rural astronomical phenomenon and the weather, biological landscape, topography, rural residents, living scenes, facility supplies, food and clothing, infrastructure, rural buildings, production tools, production scenes, and production content are twelve main categories of rural landscapes, among which two categories, infrastructure and rural buildings, are reflected in the rural living landscape and rural productive landscape. The results of this research are consistent with some existing related studies in the construction of core categories, but there are differences in the main categories and basic categories, which are due to the fact that this research paid more attention to the refinement of landscape dimensions and types. (2) The short video viewer evaluations showed that comments and preferences for rural ecological landscapes and living landscapes are frequent, especially the architectural landscapes in ecological landscapes and living landscapes, which have high visual contents and obvious visual effects. In the productive landscape, except for the category of production content, the amount of comments and preferences were quite low. As the productive landscape is mostly reflected in dynamic activities, participating is necessary to gain perception. (3) Short video viewers have a high overall valuation and preference for rural landscapes, but they raised concerns about discordant infrastructure and service facilities, potential safety hazards, sanitation, and other aspects.

### 5.2. Recommendations

**(1) Create an overall landscape.** A rural landscape is a combination of various landscape elements that are comprehensive, coordinated, and organic [55]. Artistic conception is a realm and emotion that arouses people’s imagination. The essence of rural artistic conception lies in “nostalgia” and “rural scenery”, which are integrated, coordinated, and organic. The expression of rural artistic conception requires the combination of various rural landscape elements. The evaluation and analysis results of short video viewers showed that people can use rural landscape elements to build an ecological artistic conception, with mountains and rivers, use local cultural elements to express a rural humanistic style, and use various types of plants to create the natural beauty of four distinct seasons, spring flowers and autumn leaves, highlighting the rich flavor of life through ideal rural poetic imagery [56]. At the same time, it is necessary to comprehensively consider the cultural, ecological, economic, practical, and other aspects involved and incorporate multiple factors into landscape improvement [57].

**(2) Pay attention to the expression of nostalgia**. Homesickness belongs to the category of nostalgia, which has been proven to be part of people’s psychology in the postmodern era. Home is the core focus of nostalgia [55]. The evaluation results of short video viewers showed that the rural landscape expressed in the short videos aroused the viewer’s memory and thoughts of home to the greatest extent. Natural elements such as the sun, moon, stars, and the seasonal landscape in their hometown, as well as the emotional memory of happiness and sadness [55], were the sentiments most often expressed by viewers. This is a kind of memory and yearning for the past years and situations [58] and it is a spiritual requirement for people [59]. Therefore, the expression of nostalgic elements and the protection of nostalgic heritage should be strengthened as key areas in future rural construction and the practice of rural tourism development.

**(3) Enhance tourist experiences.** In the composition of rural landscape categories and basic elements, some categories and elements have weak visual effects, and the viewer’s value perception is low, resulting in a low overall evaluation and preference. Therefore, in the construction of rural landscapes, we should strengthen the presence of participatory and experiential activities, in combination with the content of rural production and life, such as farming activities experiences and rural life experience, putting tourists into the authentic rural scene, to improve their all-round rural experience.

**(4) Strengthen resource protection.** On the one hand, the fragility of the rural ecology itself can easily lead to the destruction of the rural ecological landscape. The impact of the integration of modern cultural elements into the countryside also often causes the decline of the original rural culture, which is not conducive to the embodiment of high-quality rural natural and cultural landscapes. The evaluation results of short video viewers also showed that the above factors are the main sources of negative emotions. Therefore, we should focus on protecting the rural ecological environment and rural native culture and display the rural native landscape to the greatest extent.

### 5.3. Discussion

(1) The samples studied in this paper were TikTok short videos that reflect the rural landscape, particularly video content with a high like rate. The videos may have had common styles in expressing the content of rural landscapes, which may have limited the representativeness of the samples. (2) Although the number of middle-aged and older users is gradually increasing, young people under the age of 35 are still the majority of TikTok users. This phenomenon also makes the representativeness of the sample in this study somewhat inadequate. (3) The expression forms of short video mainly include visual effects, but do not involve the senses of hearing, taste, smell, or touch. The dubbing in the works is made by the short video publisher to enhance the effect, which may be different from the actual scene of the rural landscape expressed in the video. The perception of watching short videos and the effect of immersive experience will also be different, which will affect the analysis effect to a certain extent.

## Figures and Tables

**Figure 1 ijerph-19-10115-f001:**
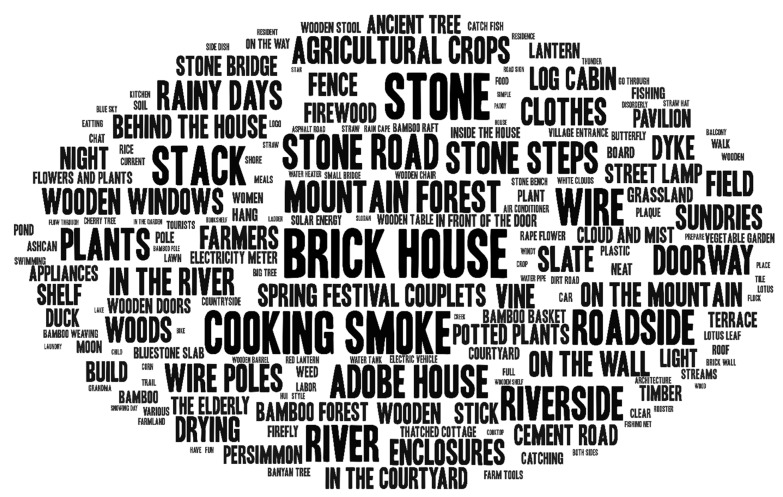
Basic elements of the rural landscape word cloud map.

**Figure 2 ijerph-19-10115-f002:**
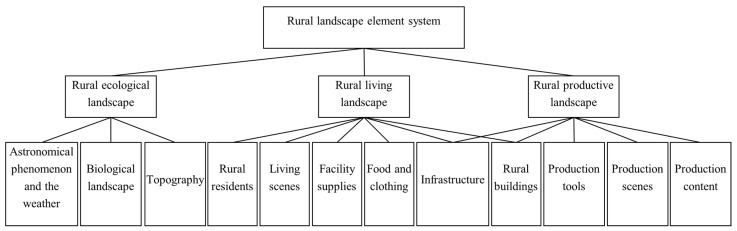
Analysis of the formation of the main rural landscape categories.

**Figure 3 ijerph-19-10115-f003:**
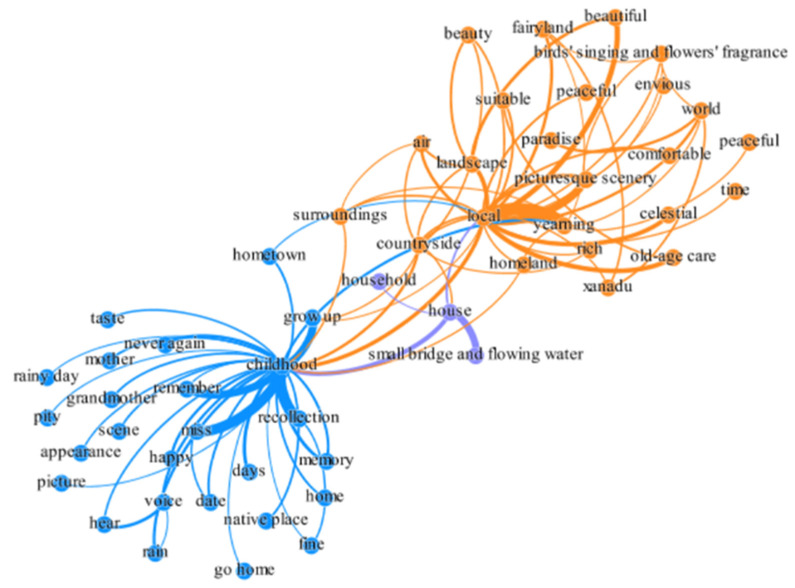
Semantic network analysis results of the rural landscape review.

**Table 1 ijerph-19-10115-t001:** Analysis of the basic characteristics of the sample videos.

Technical Parameters			Absolute Number	Ratio
**Visual elements**	Animation	Yes	38	17.1%
		None	184	82.9%
	Subtitle	Yes	12	5.4%
		None	210	94.6%
	Stability	Yes	222	100%
		No	0	0
**Auditory elements**	Music	Yes	175	78.8%
		None	47	21.2%
	Simultaneous sound	Yes	135	60.8%
		None	87	39.2%
	Narration	Yes	5	2.3%
		None	217	97.7%
**Positional information**	Yes		58	26.1%
	None		164	73.9%
**Presentation**	Full screen		218	98.2%
	Half screen		2	0.9%
	One-third screen		2	0.9%
**Duration**	Within 30 s		214	96.4%
	30–60 s		6	2.7%
	More than 60 s		2	0.9%
**Generated mode**	UGC		180	81.1%
	PGC		42	18.9%
**Likes**	1–5 w		76	34.2%
	5–10 w		44	19.8%
	More than 10 w		102	45.9%
**Comments**	Below 5000		145	65.3%
	5000–1 w		28	12.6%
	More than 1 w		49	22.1%
**Sharing**	Below 5000		128	57.7%
	5000–1 w		49	22.1%
	More than 1 w		45	20.2%

**Table 2 ijerph-19-10115-t002:** Example analysis of the formation of basic categories of a rural landscape.

Landscape Elements	Basic Category
moon, starry sky, stars, blue sky and white clouds, full moon, bright moon, night, morning…	A1 astronomical landscape
grass, banyan tree, cherry tree, persimmon tree, ancient tree, plants, bamboo…	B1 plants in a rural landscape
streams, current, lake…	C2 waterscape
pump, harvester…	D2 agricultural machinery
cattle farming, mechanized farming	E1 working in the fields
Rice, rape, corn…	F3 crops
stone bridge, stone steps, stone slab bridge, ghaut, dam…	G1 transportation facilities
yard, courtyard, kitchen, hearth, balcony, fence, railing, wooden door…	H4 auxiliary space of rural buildings
middle-aged women, grandmother, the elderly man, angler…	I1 rural residents
washing clothes, washing vegetables, cooking…	J1 life activities
plastic bucket, bamboo crate, bamboo basket	K1 life utensils
coir raincoat, clothes, straw hat, bamboo hat…	L2 clothes
…	…
A total of 252 rural landscape elements are defined	A total of 32 basic categories are formed

**Table 3 ijerph-19-10115-t003:** High-frequency characteristic words for rural landscape evaluation, TOP50.

Rank	High-Frequency Words	Frequency	Rank	High-Frequency Words	Frequency
1	childhood	17,342	26	xanadu	1870
2	local	16,216	27	hometown	1759
3	world	11,022	28	hear	1743
4	fairyland	9044	29	infancy	1726
5	yearning	5349	30	maturity	1619
6	countryside	4916	31	suitable	1614
7	picturesque scenery	3514	32	celestial	1593
8	scenery	3408	33	day	1576
9	miss	3286	34	rain	1574
10	young	3205	35	birthplace	1456
11	recollection	3180	36	time	1423
12	grow up	3061	37	rich	1392
13	house	2579	38	rice	1385
14	remember	2476	39	real	1385
15	air	2390	40	fine	1380
16	beautiful	2348	41	paradise	1376
17	birds’ singing and flowers’ fragrance	2212	42	happy	1324
18	smoke	2188	43	peaceful	1317
19	sound	2103	44	comfortable	1230
20	household	2095	45	home	1220
21	small bridges flowing water	2095	46	native place	1211
22	environment	2040	47	grandmother	1206
23	old-age care	2039	48	mother	1205
24	wonderful	1982	49	scene	1203
25	memory	1885	50	feeling	1094

**Table 4 ijerph-19-10115-t004:** Sentiment analysis results of the rural landscape review.

Emotional Attitude	Ratio	Intensity	Ratio
Positive	46.96%	High degree (above 20)	6.12%
		Medium (10–20)	13.54%
		General (0–10)	27.30%
Neutral	40.70%	-	40.70%
Negative	12.34%	High degree (below minus 20)	0.21%
		Medium (−20–−10)	1.94%
		General (−10–0)	10.19%

## Data Availability

The data used to support the findings of this study are available from the corresponding author upon request (wangmin@stu.ahau.edu.cn).

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
