# Peer review of "Research on Rural Landscape Preference Based on TikTok Short Video Content and User Comments"

_ijerph, 2022, doi:10.3390/ijerph191610115_

Round 1
Reviewer 1 Report
The authors present an original approach to the study of TikTok content, analysing the way rural landscapes are presented on TikTok, as well as evaluated and commented on by users.
The study fails in addressing ethical concerns regarding the use of public social media data for scientific purposes. After all, it (a) does not indicate that users were asked/informed about their data being used for research, and (b) presents a possible blueprint for the analysis of large amounts of visual TikTok data and user comments. If the steps done by researchers in this study, were done by AI instead, a similar approach could be used for social surveillance. Maybe the authors can address this concern and clarify if and if so, how users were asked whether they wanted their content being used in this study.
The study provides much information on the way data was collected and on the process of data coding, emphasising that a Grounded Theory approach was applied. Yet, the authors do not refer to any specific literature on Grounded Theory, which makes it hard to comprehend how the authors treated their materials as ontological entities. Information on the specific Grounded Theory approach would also help in better understanding the epistemology behind the construction of basic, main and core categories.
While the article's strength lies in its minutae description of the research process and coding strategy, it misses to ask the question who uses (and comments on) TikTok. While it is true that TikTok is very popular and widely used, it still is a social media platform with a great share of younger users and comparably fewer older users. This limitation alongside others (e.g. barriers to TikTok use due to a lack of material resources, no access to the internet, or the technological devices, etc.) should be mentioned since the aim of the paper aims at describing a popular understanding of rural landscapes and the evaluation of said landscapes.
Tables should be formatted to improve their readability (e.g. by making some use of vertical and/or horizontal lines or, alternatively extra spacings). In general the tables and graphs are very interresting and provide very valuable information.
Before publishing I strongly recommend English proof reading (I highlighted some mistakes in the document attached, however, there are more errors still).
In sum, I consider this paper to present an original an innovative approach to the study of TikTok content, with valuable insights for the popular perception of landscape imagery on social media and interresting recommendations for the human custodianship of rural landscapes.
Reviewer 2 Report
This manuscript mined TikTok video content from 222 short videos to analyse rural landscape preferences by viewers. The authors suggest the need to address rural artistic conception and the expression of nostalgia to enhance tourism experience in rural landscapes to strengthen the protection of natural and cultural landscapes.
This manuscript is generally well written and interesting with innovative and cross-correlating methods that expose values in the landscape preferences of the data they use. There is a potential for the results to provide useful application in rural landuse policy through re-framing the research so that it reports on the results of the analysis and avoids the second-order judgements that compromise the discussion and conclusions.
A number of issues in this manuscript pertain to: (1) the literature on which they study draws; (2) elements of the description of methodology and methods; (3) tables and figures; (4) Evidence for discussion points and conclusions; (5) English language expression.
(1) Literature:
a. Firstly, too few recent references relate to the theory and practice of landscape values. This is both in the initial literature review, and in the discussion of the results.
b. There are two disciplinary areas that are neglected in this research: Rural Sociology and Landscape Architecture. Rural Sociology investigates and exposes the discord and contradiction of rural production and rural romanticism, and the consequences for rural people. Tourism has a price for rural activities, particularly in production landscapes. This manuscript does not address these contradictions, and the consequences for agricultural production. Landscape Architecture has extensive commentary of structure and function in landscapes, with particular importance for rural policy and landscape planning.
c. Extensive PPGIS research uses a typology of landscape values that should be discussed and potentially applied here (see Brown, Reed et al; Ernoul et al), rather than re-inventing the wheel through Grounded Theory approaches. Much of this has been tried and tested, and there is no reason why these theories should not be re-tested to show latent and improved classifications and characterisations, particularly in the settings and context of this research. Commentary of the context of ethnicity and geography would make a very useful addition to the international literature in this field.
d. There are some places that referencing has not been used that is needed. Examples include the following but the manuscript needs a thorough review to address this issue: section 3.3 “Production-Living-Ecological” concept is the most widely used rural composition dimension the field of rural research; section 5.2 “The essence of rural artistic conception lies in “nostalgia” and “rural scenery”, which is integrated, coordinated and organic”; Section 5.2 Point (2) “Home is the core intention of nostalgia”.
e. The sources of literature in the para starting “Relevant research at home and abroad….” Show a selective cultural bias, as well as a geographical bias. This limits the applicability of the research in current form.
f. In addition to point (e) many of the statements are overarching and generalising but based on only one reference. Each of these statements requires more than one reference to demonstrate a thorough referencing of the broader literature that this research draws on.
(2) Methodology and Methods
a. A very good overview of landscape visual assessment techniques but not clear why the “direct methods” were not better outlined and critiqued, thus it is not obvious why or actually if, indirect methods or a hybrid was used. In addition, this section needs clarification such as providing a clearer definition which includes definition and provides detail on the meaning for “viewer’s visual effects”. This is also an issue in the abstract where it is not clear what “most operational scale” means, represents, or applies to.
b. There are limitations to this study which include the sources of material (or data) which no doubt reflect specific socio-demographic sections of society. The methods section must include a description (at minimum age, gender and ethnicity) that accounts of these limitations in the discussion.
c. Section 3.1 states that: “This study follows the principle of objectivity”. Objectivity and the notion of “truthfully” used in the next sentence, are always a misnomer in research that develops categories and classifications. Subjectivity is essential in this process. It is worth noting the distinction between inductive and deductive research here as a way of addressing the conventional biases faced by qualitative researchers applying quantitative methods. It is worth saying how the method was applied and why without the binary claims of truth and objectivity which are misnomers. In addition to this, the authors need to take care that their personal judgements are not invested in the results if their objectivity claims are to be accommodated. There are numerous occurrences of value-investment that translate to second-order judgements moving conclusions beyond the supporting evidence. Examples are in the way in which results are reported with concepts such as “positive emotions” and “negative emotions”, “discordant infrastructure” ; Section 4.4 “the proportion of high negative emotions is [EXTREMELY] low”
d. In numerous places the manuscript (and research) depends on classifications and characterisations which are not adequately defined or described, for example: 3.2 “The [BASIC] categories” also referred to again in Table 3; “high-quality rural landscapes”. This also applies to the concept of “core categories” in section 3.3; “[HIGH-QUALITY] rural landscapes”.
e. Details of methods need to be addressed which includes: the actual number of “repeated analysis and comparison” mentioned in 3.1; how “the TXT data text in ANSI format is [SORTED OUT]”; “Residents’ [ATTACHMENT] to the local rural landscape”.
(3) Tables & Figures
a. In Table 1 the categories don’t nest visually so that Music appears to be part of the Visual Elements Super Category etc etc. Re-formatting for clarity here is essential.
b. Figure 1 is only useful if correlated with a quantified form. In addition, the content of the figure must be clearly represented so that not only the larger text is readable, but also the smallest text.
c. Table 2 and Table 3 must appear on one page so that the category headers are linked to the data presented.
d. Table 3 is better represented in Figure 2, thus making Table 3 redundant.
(4) Evidence for discussion points and conclusions
a. no refs or evidence to support these statements: Section 4.5.2 “Rural life carries more rural history and culture, is the core embodiment of rural memory, and is an important component of rural cultural landscape”; “It can be seen that the rural life landscape triggers the local complex of the viewer, produces…..which is the core meaning of the rural landscape”.
b. Section 5.2 “rural artistic conception” is a concept introduced in the final section of the manuscript with inadequate definition or review, and no operationalisation in the methods or analysis…this needs to be addressed.
(5) English language expression
a. Sexist language must be avoided such as Section 1 Introduction “man-land relationship”.
b. Informal language lacks clarity eg Section 2 “All the [SHOTS] in the visual elements”.
c. Some categories need a re-think for their common use English meaning such as: “astronomical climate” which is usually associated with solar weather systems, rather that weather at the landscape level. It would be useful to differentiate between weather and climate here; the meaning of “ecological landscapes” might be clearer if associated with nature and the natural environment, rather than the science of ecology and functioning of natural systems; “the [GENUS] of mulberry and bamboo”;
d. A re-wording for clarity is required for: “it needs to participate in the experience to gain perception”.
Round 2
Reviewer 2 Report
Dear Authors, I found the content interesting, and the methods innovative and constructive. In general the comments I provided in the review have been addressed. With two minor comments which follow that still need to be addressed before the research is ready for publication.
Two review comments have not yet been adequately addressed, as follows:
Point (4) Evidence for discussion points and conclusions: b.Section 5.2 "rural artistic conception" is a concept introduced in the final section of the
manuscript with inadequate definition or review, and no operationalisation in the methods or analysis…this needs to be addressed.
and,
Point (5) English language expression: c. Some categories need a re-think for their common use English meaning such as: "astronomical climate" which is usually associated with solar weather systems, rather than weather at the landscape level. It would be useful to differentiate between weather and climate here...
Again, great to see this kind of interesting research reported, and adding to the international literature on rural image and landscape valuing.
